# AMPK Activation Is Important for the Preservation of Insulin Sensitivity in Visceral, but Not in Subcutaneous Adipose Tissue of Postnatally Overfed Rat Model of Polycystic Ovary Syndrome

**DOI:** 10.3390/ijms23168942

**Published:** 2022-08-11

**Authors:** Bojana Mićić, Ana Teofilović, Ana Djordjevic, Nataša Veličković, Djuro Macut, Danijela Vojnović Milutinović

**Affiliations:** 1Department of Biochemistry, Institute for Biological Research “Siniša Stanković”, National Institute of Republic of Serbia, University of Belgrade, 142 Despot Stefan Blvd, 11060 Belgrade, Serbia; 2Clinic for Endocrinology, Diabetes and Metabolic Diseases University Clinical Centre of Serbia, Faculty of Medicine, University of Belgrade, Doktora Subotića 13, 11000 Belgrade, Serbia

**Keywords:** polycystic ovary syndrome, obesity, visceral adipose tissue, subcutaneous adipose tissue, insulin resistance, 5α-dihydrotestosterone, early postnatal overfeeding, lipogenesis, lipolysis, AMP-activated kinase

## Abstract

Polycystic ovary syndrome (PCOS) is a well-known reproductive syndrome usually associated with obesity, insulin resistance, and hyperinsulinemia. Although the first signs of PCOS begin early in adolescence, it is underexplored whether peripubertal obesity predisposes women to PCOS metabolic disturbances. To highlight that, we examined the impact of postnatal overfeeding-induced obesity, achieved by litter size reduction during the suckling period, on metabolic disturbances associated with visceral and subcutaneous adipose tissue (VAT and SAT) function in the 5α-dihydrotestosterone (5α-DHT)-induced animal model of PCOS. We analyzed markers of insulin signaling, lipid metabolism, and energy sensing in the VAT and SAT. Our results showed that postnatally overfed DHT-treated Wistar rats had increased VAT mass with hypertrophic adipocytes, together with hyperinsulinemia and increased HOMA index. In the VAT of these animals, insulin signaling remained unchanged while lipogenic markers decreased, which was accompanied by increased AMPK activation. In the SAT of the same animals, markers of lipogenesis and lipolysis increased, while the activity of AMPK decreased. Taken together, obtained results showed that postnatal overfeeding predisposes development of PCOS systemic insulin resistance, most likely as a result of worsened metabolic function of SAT, while VAT preserved its tissue insulin sensitivity through increased activity of AMPK.

## 1. Introduction

Polycystic ovary syndrome (PCOS) is one of the most common endocrinopathies among reproductive-aged women, with a prevalence ranging from 5.5 to 19.9%, depending on what diagnostic criteria are used [1,2]. It is a complex condition, which in addition to the reproductive implications such as anovulation, hyperandrogenism, and characteristic ovarian morphology, impacts a broad range of metabolically active tissues and organs [3]. Although the precise etiology of PCOS has not been elucidated so far, the pivotal role of androgens is undeniable [4]. In addition, obesity and insulin resistance have a prominent role in the pathogenesis of the syndrome [5,6].

It has been recognized that nutritional conditions in early life play a significant role in the development of overweight and related chronic comorbidities in adulthood [7]. Nutritional status during the perinatal period can affect appetite and growth dynamics and provoke metabolic malprogramming, making overnourished individuals more susceptible to overweight, obesity, and related comorbidities, including reproductive ones [7,8]. The first manifestations of PCOS begin in early adolescence, and numerous studies have shown that obesity during this period promotes the development of pubertal and adolescent PCOS [9,10,11]. This indicates that peripuberty is a critical developmental window when the pathophysiology of PCOS widens.

Androgens are associated with type 2 diabetes (T2DM) in a sex-specific manner, and androgen excess in females is recognized as a risk factor for the development of T2DM, which becomes particularly evident in conditions such as PCOS [12]. The odds of developing metabolic disturbances, including impaired glucose tolerance, T2DM, and metabolic syndrome, are two to four times greater in women with PCOS compared to age and BMI-matched controls [13]. It is well documented that up to 80% of women with PCOS have impaired insulin sensitivity, and as a consequence, insulin resistance and hyperinsulinemia, contributing to the maintenance of hyperandrogenemia and to the development of cardiometabolic diseases in those women [14]. Impairment of insulin sensitivity in metabolically active tissues depends on the alteration of key components of the insulin pathway such as insulin receptor substrate 1 (IRS-1) and protein kinase B (Akt), as well as their phosphorylation status.

White adipose tissue plays a crucial role in the maintenance of systemic energy homeostasis through its endocrine function and energy storage [15]. Depending on the localization and functions, visceral and subcutaneous adipose tissue (VAT and SAT, respectively) can be distinguished. Changes in the composition and function of different adipose tissue depots are important for understanding the mechanisms underlying obesity and consequential pathologies. Pathogenic expansion of the VAT is related to insulin resistance, inflammation, and dyslipidemia, while the expansion of SAT is frequently associated with the improvement of metabolic status and insulin sensitivity [16].

White adipose tissue is the main organ for the storage and mobilization of energy in the form of triglycerides, as well as for the control of circulating free fatty acids (FFA). Lipolysis is the main process of the hydrolysis of lipids by the enzymes adipose tissue triglyceride lipase and hormone sensitive lipase (HSL), while lipid synthesis involves lipoprotein lipase (LPL), acetyl-CoA carboxylase (ACC), and fatty acid synthase (FAS). Adipose tissue, along with skeletal muscles, is the main metabolic target of androgens. Androgens are known to inhibit proliferation and differentiation of preadipocytes [17,18]. Consequential hypertrophy of differentiated adipocytes, as a compensatory mechanism to restricted hyperplasia, impairs adipocyte function and insulin sensitivity. Androgens also exert their effect through the regulation of the central nervous system, contributing to the development of abdominal obesity, insulin resistance, and T2DM in hyperandrogenic females [12].

AMP-activated kinase (AMPK) is an energy sensor of the cell that regulates key proteins involved in carbohydrate and lipid metabolism. AMPK activation results in the suppression of anabolic pathways and stimulation of ATP production through fatty acid oxidation, muscle glucose transport, and caloric intake [19]. At the adipocyte level, activated AMPK blocks the expression of late adipogenic markers such as FAS and the transcription factors involved in preadipocyte differentiation [20]. AMPK is also associated with the decreased transcription rate of lipogenic enzymes and with increased activity of enzymes involved in lipolysis. Furthermore, activated AMPK in adipocytes promotes fatty acid oxidation [21,22].

Despite the association of PCOS with obesity, there is little evidence suggesting how dietary factors or obesity predispose women to PCOS metabolic disturbances. We hypothesized that the increased caloric intake in the early postnatal period will aggravate metabolic disturbances, specifically associated with VAT and SAT functions, in the PCOS animal model induced by continuous treatment with 5α-dihydrotestosterone (5α-DHT). To investigate that, a well-established Wistar rat model previously shown to manifest both metabolic and reproductive features of PCOS [23,24,25] has been upgraded with early postnatal litter size reduction. Litter size reduction has been proven to be a useful experimental approach to study the effects of early overnutrition and consequential overweight in youths and adults [7]. Taking these into account, in the present study we analyzed parameters of systemic insulin sensitivity, together with insulin signaling, lipid metabolism, and energy sensing in both VAT and SAT of postnatally overfed female Wistar rats chronically treated with 5α-DHT.

## 2. Results

### 2.1. Physiological and Biochemical Parameters

Weight gain dynamics during the suckling period of female rats raised in small (SL) and normal (NL) litters is shown in Figure 1. Animals raised in small litters achieved significantly higher body mass than animals raised in normal litters from the 8th day of age until the weaning.

After pellet implementation and a 90-day treatment, we analyzed the effects of litter size manipulation, DHT treatment, and their combination on physiological parameters. *Two-way* ANOVA showed the main effect of litter size reduction on energy intake throughout life (F (1, 56) = 14.31, *p* < 0.001). Body mass was affected by litter size reduction (F (1, 20) = 13.48, *p* < 0.001) and DHT treatment (F (1, 20) = 5.50, *p* < 0.05), while only litter size reduction influenced absolute and relative masses of the VAT (F (1, 20) = 21.61, *p* < 0.001; F (1, 20) = 19.48, *p* < 0.001, respectively). At the same time, none of the two treatments affected the absolute and relative-to-body masses of the SAT in the analyzed groups (Table 1).

All DHT-treated animals were acyclic, in the diestrus phase of the estrous cycle, while placebo animals preserved normal cyclicity, with a 4–5 day estrous cycle. *Two-way* ANOVA revealed the significant effect of DHT treatment on absolute masses of ovaries (F (1, 20) = 23.99, *p* < 0.001) and uteri (F (1, 20) = 24.14, *p* < 0.001), as well as their relative-to-body ratios (F (1, 20) = 29.24, *p* < 0.001; F (1, 20) = 28.11, *p* < 0.001, respectively).

Neither litter size reduction nor DHT treatment influenced blood triglycerides, FFA or glucose levels (Table 2). However, parameters of insulin sensitivity were impaired. *Two-way* ANOVA indicated significant effect of litter size reduction (F (1, 20) = 22.13, *p* < 0.001), DHT-treatment (F (1, 20) = 8.86, *p* < 0.01), and their interaction (F (1, 20) = 4.58, *p* < 0.05; *post hoc* test: *** *p* < 0.001 SL-DHT vs. NL-Placebo; ^###^
*p* < 0.001 SL-DHT vs. NL-DHT, ^++^ *p* < 0.01 SL-DHT vs. SL-Placebo; Table 2) on blood insulin level. In line with this, homeostasis model assessment (HOMA) index was also affected by both the treatments and their interaction (litter size: F (1, 20) = 21.55, *p* < 0.001; DHT-treatment: F (1, 20) = 7.82, *p* < 0.05; interaction: F (1, 20) = 5.07, *p* < 0.05), so that it was significantly increased in DHT-treated animals from small litters compared to all other groups (*** *p* < 0.001 SL-DHT vs. NL-Placebo; ^###^ *p* < 0.001 SL-DHT vs. NL-DHT, ^++^ *p* < 0.01 SL-DHT vs. SL-Placebo; Table 2). On the other hand, only litter size reduction affected intraperitoneal glucose tolerance test (ipGTT) area under curve (AUC) (F (1, 17) = 17.24, *p* < 0.001).

### 2.2. Histological and Morphometric Analysis of VAT and SAT

Representative sections of VAT are shown in Figure 2A. According to *two-way* ANOVA analysis, litter size reduction and DHT treatment affected both diameter (litter size: F (1, 20) = 34.50, *p* < 0.001; DHT-treatment: F (1, 20) = 6.19, *p* < 0.05) and area (litter size: F (1, 20) = 38.16, *p* < 0.001; DHT-treatment: F (1, 20) = 7.64, *p* < 0.05) of adipocytes in the VAT. Representative sections of SAT are shown in Figure 2D. Unlike the changes observed in the VAT, none of the treatments affected either diameter (Figure 2E) or area (Figure 2F) of adipocytes in the SAT.

### 2.3. Insulin Signaling in the VAT

The VAT insulin sensitivity was estimated by the protein level of IRS-1 with its inhibitory phosphorylation on Ser^307^, Akt with its stimulatory phosphorylation on Ser^473^, and total extracellular signal-regulated protein kinase 1/2 (Erk1/2) with its stimulatory phosphorylation on Thr^202^ and Tyr^204^. As shown in Figure 3, none of the analyzed kinases involved in insulin signaling pathways have been affected by the treatments or their interaction.

### 2.4. Markers of Lipid Metabolism in the VAT and SAT

Litter size reduction and DHT-treatment alone, as well as their interaction, affected markers of lipid metabolism in the VAT. Namely, *two-way* ANOVA detected significant effect of litter size reduction, DHT treatment and their interaction on the expression of the genes encoding crucial enzymes involved in *de novo* lipogenesis: FAS (litter size: F (1, 20) = 21.69, *p* < 0.001; DHT treatment: F (1, 20) = 9.45, *p* < 0.01; litter size × DHT treatment: F (1, 20) = 6.57, *p* < 0.05), ACC (litter size: F (1, 20) = 12.93, *p* < 0.01; DHT treatment: F (1, 20) = 8.70, *p* < 0.01, litter size × DHT treatment: F (1, 20) = 13.35, *p* < 0.01), as well as on stearoyl-CoA desaturase 1 (SCD1) gene expression (litter size: F (1, 20) = 17.10, *p* < 0.001; DHT treatment: F (1, 20) = 10.37, *p* < 0.01, litter size × DHT treatment: F (1, 20) = 9.82, *p* < 0.01). As shown in Figure 4A, a *post hoc* test revealed that relative expression of FAS, ACC, and SCD1 encoding genes was significantly lower in all treated groups compared to normal litter placebos (*** *p* < 0.001 NL-DHT, SL-Placebo and SL-DHT vs. NL-Placebo). Figure 4B shows relative expression of LPL and phosphoenolpyruvate carboxykinase (PEPCK) encoding genes that are involved in triglyceride synthesis from circulating fatty acids and their storage in adipocytes. Only the interaction of the treatments affected the expression of the gene encoding PEPCK (F (1, 20) = 5.89, *p* < 0.01), with no difference detected by the *post hoc* test. The expression of the LPL gene remained unaltered.

Simultaneously, litter size reduction and its interaction with DHT treatment affected the expression of the HSL encoding gene (litter size: F (1, 20) = 8.36, *p* < 0.01; litter size × DHT treatment: F (1, 20) = 13.50, *p* < 0.01) in the VAT. A further *post hoc* test showed that HSL encoding gene expression was lower in the NL-DHT and SL-Placebo group in comparison with NL-Placebo (* *p* < 0.05, *** *p* < 0.001, Figure 4C).

In the SAT, the relative expression of genes encoding enzymes involved in the process of lipogenesis: ACC (F (1, 20) = 16.54, *p* < 0.001)*,* LPL (F (1, 20) = 35.33, *p* < 0.001), and PEPCK (F (1, 20) = 29.14, *p* < 0.001), was significantly affected by the DHT treatment, as reveled by *two-way* ANOVA.

The *two-way* ANOVA showed significant effect of DHT treatment (F (1, 20) = 27.93, *p* < 0.001), as well as significant interaction between litter size reduction and DHT treatment (F (1, 20) = 8.89, *p* < 0.01) on the expression of HSL encoding gene, a lipolysis marker, in the SAT. A *post hoc* test revealed that HSL gene expression increased in DHT treated animals from small litters as compared to both small and normal litter-raised placebos (^+++^
*p* < 0.001, ** *p* < 0.01, respectively, Figure 5B).

### 2.5. Energy Sensing in the VAT and SAT

Although AMPK protein expression in the VAT was affected by neither early postnatal overfeeding and DHT treatment nor their combination, we have noted a significant effect of DHT treatment on the pAMPK-Thr^172^ expression and pAMPK-Thr^172^/AMPK ratio (F (1, 20) = 7.06, *p* < 0.05; F (1, 20) = 21.84, *p* < 0.001, respectively) in this tissue (Figure 6). It is also noteworthy that the interaction between DHT treatment and litter size reduction is on the brink of significance for pAMPK-Thr172/AMPK ratio (F (1, 20) = 3.87, *p* = 0.06).

In the SAT, the *two-way* ANOVA detected a significant effect of litter size reduction on the pAMPK-Thr^172^ expression (F (1, 20) = 84.87, *p* < 0.001, as well as on the expression of total AMPK (F (1, 20) = 166.2; *p* < 0.001) (Figure 7).

## 3. Discussion

PCOS is a complex condition and, despite substantial research, the underlying mechanisms and etiopathogenesis of the syndrome remain elusive. In addition to the reproductive implications, such as anovulation, infertility, and hyperandrogenism, metabolic disturbances such as obesity, dyslipidemia, and insulin resistance are also hallmarks of this disorder [3]. Treatment of female rats with nonaromatizable DHT produces a hyperandrogenic animal model of PCOS, which is a well-established model that exhibits both reproductive and metabolic characteristics of the syndrome [23,26].

PCOS symptoms are the most pronounced in women of reproductive age, but the disorder also carries the risk for metabolic and cardiovascular comorbidities in prepuberty and adolescence [2]. Since the first signs of PCOS appearance in young prepubertal girls are often associated with obesity [9,10,11], we upgraded the PCOS animal model in this study with the introduction of increased caloric intake in the early postnatal period. This was done in order to analyze the relationship between peripubertal obesity and hyperandrogenemia in the development of PCOS. Namely, on the second *postpartum* day, litter size was manipulated to form small and normal litters, that is, to provoke early postnatal overfeeding, which has been proven to lead to weight gain and metabolic and nutritional malprograming [7]. Raising rats in small litters reduces competition for milk and ensures increased food consumption. In addition, some studies show that lipid content of the milk, especially triglycerides, is higher in small litter-nursing dams than in normal litter-nursing ones [27]. Indeed, in the present study, early postnatal overfeeding achieved through litter size reduction led to the increase in the body mass of small litter-raised animals in comparison with normal litter-raised ones during the suckling period. Postnatal overfeeding applied in this study possibly affected appetite control of these rats, so that they maintained increased energy intake and higher body masses even in the adult period. However, we did not record any change in blood concentrations of FFA and triglycerides, and similar findings were previously reported in other mouse and rat models [23,28]. In addition, the absolute and relative-to-body VAT masses were also positively influenced by the litter size reduction, which is in accordance with findings from other authors [7,29,30].

Both visceral adiposity and hyperandrogenemia are well-documented insulin signaling impairment factors [12]. Although the mechanisms of insulin resistance in PCOS are not fully understood, it is clear that obesity plays a crucial role in the deterioration of metabolic features of PCOS patients [31]. In the present study, fasting blood glucose levels remained unchanged, while ipGTT AUC values were affected only by litter size. The contribution of postnatal overfeeding to metabolic features of PCOS is evidenced by an elevated HOMA index and increased fasting insulin, demonstrating decreased insulin sensitivity only after combined treatments. These results indicate that the combination of these factors particularly aggravates insulin sensitivity at the systemic level and leads towards compensatory hyperinsulinemia in order to maintain normal plasma glucose levels. Systemic insulin resistance can be a consequence of disturbed insulin signaling in different metabolically active organs such as the liver, skeletal muscles, and adipose tissue. The histological analysis of VAT in the present study showed that both litter size reduction and DHT treatment contributed to the hypertrophic morphology of the adipocytes. Since there are several possible mechanisms linking visceral fat accumulation with insulin resistance [32], we analyzed protein levels of IRS-1 and its inhibitory phosphorylation on Ser^307^, as well as levels of total and phosphorylated forms of Akt and Erk1/2 kinases in this tissue. The results demonstrated that local insulin signaling in the VAT was not disturbed by the combination of postnatal overfeeding and DHT treatment. We presume that the observed disturbance of insulin sensitivity at the systemic level is not a result of insulin signaling impartment in the VAT, but rather the consequence of local insulin resistance in other metabolically active tissues, such as skeletal muscles or liver. In line with the observation of enlarged VAT adipocytes, we also assessed the expression of main lipogenic enzymes FAS, ACC, and SCD1, and found them to be significantly reduced. Such findings could suggest that, in the condition of existing obesity, lipogenic capacity of adipocytes is downregulated in an adaptive inhibitory feedback process with the aim to limit further development of fat mass. Namely, Ortega et al. [33] previously showed that FAS and ACC gene expression was downregulated in the visceral fat from overweight and obese subjects, while SCD1 inhibition in extrahepatic tissues was shown to be protective against obesity and insulin resistance in mice on a high-fat diet [34]. Taken together, it seems that VAT adipocytes from DHT-treated small-litter animals have restrained *de novo* synthesis of fatty acids, thus preserving tissue insulin sensitivity. It is well-documented that activation of AMPK, a major cellular energy sensor and a master regulator of metabolic homeostasis in adipocytes, leads to inhibition of fatty acid synthesis and lipolysis through suppression of the ACC and HSL [19]. Indeed, our results showed that the ratio of phosphorylated AMPK at Thr^172^ to total AMPK was higher in the VAT of DHT-treated animals. In line with our DHT treatment, Mitsuhashi et al. [35] showed that 3T3-L1 cells treated with testosterone had enhanced AMPK-Thr^172^ phosphorylation via the LKB1 signaling pathway and unchanged Akt phosphorylation. In addition, AMPK phosphorylates and inactivates SREBP-1c, thereby indirectly decreasing the expression of lipogenic genes ACC, FAS, and SCD1 [36], which is consistent with the results observed in this study. Taken together, it is reasonable to presume that the activation of AMPK could be the main factor contributing to the preserved VAT insulin sensitivity in the condition of disturbed systemic insulin sensitivity, through suppression of lipogenesis and restriction of lipolysis in this adipose tissue depot.

In obesity, the expansion of different depots of adipose tissues can result from adipocyte hyperplasia or hypertrophy. SAT possesses higher capacity for the formation of new adipocytes, while VAT demonstrates higher potential for hypertrophy. SAT has a “buffering role” in obesity by limiting the negative consequences of ectopic fat deposition through the adipogenesis and long-term fatty acids accumulation. This correlates with decreased risk of glucose and insulin abnormalities [32]. In the condition of caloric overload, energy is primarily stored in the SAT, and when this depot is dysfunctional, VAT can serve as an alternative energy depot [37]. In the present study, there were no changes in the SAT adipocyte diameter or number after treatments, while VAT demonstrated hypertrophic morphology in small-litter-raised animals, especially prominent after combined treatment. Similarly, studies in female non-human primates showed that a combination of hyperandrogenemia and a high-energy diet accelerated development of obesity and VAT hypertrophy, but without significant effect on SAT hypertrophy [38,39,40]. Generally, failure of adipocyte proliferation and differentiation of SAT in the state of increased caloric overload is considered a possible mechanism of its impaired functionality [41]. At the same time, perturbations of lipid metabolism were observed in this adipose tissue depot, since mRNA levels of PEPCK and LPL, together with the expression of lipolytic enzyme HSL were upregulated, resulting in higher SAT *de novo* lipogenesis and impaired insulin-driven suppression of lipolysis. Increased SAT expression of genes involved in both the synthesis and hydrolysis of fatty acids is closely associated with insulin resistance in obese women. This implies that dysfunction of SAT in overfed DHT treated animals is closely related to both visceral obesity and systemic insulin resistance. In parallel with impaired SAT, fatty acid cycling significantly lowered AMPK phosphorylation and total protein level in postnatally overfed rats regardless of DHT, which is in accordance with the study that demonstrated decreased AMPK activity and increased *de novo* lipogenesis in the dysfunctional SAT of insulin-resistant obese women [42].

Our results showed that postnatal overfeeding could be a critical factor that leads to decreased systemic insulin sensitivity and consequent hyperinsulinemia in the animal model of PCOS, still not accompanied by impaired insulin signaling in the VAT. Although postnatal overfeeding and androgen excess induced VAT adipocytes hypertrophy, the increased activity of AMPK in this depot most likely contributed to the maintenance of the local insulin sensitivity through restrained lipogenesis. On the other hand, in the SAT, activity of AMPK was decreased by overfeeding, while the lipogenic and lipolytic factors were increased by DHT treatment. Simultaneously attenuated AMPK activity and impaired fatty acid cycling in the SAT of postnatally overfed 5α-DHT treated animals point to its dysfunction, which most likely contributes to the observed hypertrophy of VAT and decreased systemic insulin sensitivity in these animals. The highlight of this study is the introduction of an upgraded animal model of PCOS that contributes to a better understanding of the pathophysiology of this syndrome, especially in the view of synergy between obesity and insulin resistance. Therefore, our upgraded animal model represents a valuable tool for further development of new therapeutics needed for the treatment of this complex disease.

## 4. Materials and Methods

### 4.1. Experimental Design and Animals

Female Wistar rats were two days old at the beginning of the study. On the second day after birth, litter size was adjusted to form small litters (SL) and normal litters (NL) that contained three pups and ten pups with a lactating dam, respectively. Litter size was manipulated to induce early postnatal overfeeding in the SL group.

On the 22nd day after birth, pups were separated from lactating dams and implanted subcutaneously in the neck region with pellets containing 5α-DHT or placebo pellets lacking bioactive substance (Innovative Research of America, Sarasota, FL, USA). Four experimental groups were formed (*n* = 6 animals per group): animals from normal litters implanted with placebo pellets (NL-Placebo); animals from normal litters implanted with pellets containing 7.5 mg of 5α-DHT and releasing 83 µg per day for 90 days (NL-DHT); and corresponding two groups with animals from small litters (SL-Placebo and SL-DHT). The dose of 83 µg of DHT per day was selected to correspond to the increase in androgen concentrations in the blood observed in women with PCOS [23,43].

After weaning and pellet implantation, all animals were housed three per cage, under controlled temperature (22 ± 2 °C), constant humidity, and standard 12 h light/dark cycle (lights on at 7:00 a.m.), with *ad libitum* access to standard laboratory chow (Veterinary Institute “Subotica”, Serbia) and tap water. Food intake and body mass were monitored throughout the treatment, and energy intake was calculated using the formula: Energy intake = mass of the food ingested per day per cage (g) × 11 kJ.

At the end of 90 days of treatment, animals were sacrificed by rapid decapitation with a guillotine (Harvard Apparatus, Holliston, MA, USA) in the diestrus phase of the estrus cycle, after 6 h of fasting.

All the experiments and animal treatments were done following the guidelines of the Serbian Laboratory Animal Science Association (SLASA) and EEC Directive 2010/63/EU for the protection of animals used for experimental and other scientific purposes. The study was approved by the Ethical Committee for the Use of Laboratory Animals of the Institute for Biological Research “Siniša Stanković”, University of Belgrade (No. 01-02/19).

### 4.2. Serum and Tissue Preparation

Trunk blood was collected at decapitation and incubated for 40 min at room temperature to clot. Glucose levels were measured at decapitation, from the whole blood, by Accu-Chek strips (Roche, Mannheim, Germany). Serum was separated by low-speed centrifugation (1600× *g*, 10 min, 14 °C) and stored at −70 °C for subsequent analysis.

Immediately after decapitation, VAT and SAT were carefully excised [44], washed in 0.9% saline, dried, and weighted. Isolated VAT (retroperitoneal and perirenal depots) and SAT (anterior and posterior depots) were divided into parts for Western blot, qPCR, and histological analyses. Parts of tissues for molecular analyses were snap-frozen in liquid nitrogen and kept at −70 °C until protein and RNA isolation. Parts of VAT and SAT for histological analysis were fixed in 4% paraformaldehyde.

### 4.3. Determination of Blood Parameters

Serum insulin concentrations were measured commercially by the RIA method (RIA-Insulin (PEG), # 120; INEP, Belgrade, Serbia). Assay sensitivity was 0.6 mIU/L, intra-assay coefficient of variation (CV) was 2.5% and inter-assay CV was 7.7%.

The levels of FFAs and triglycerides were determined by a Randox NEFA kit (Randox Laboratories Ltd., Crumlin, UK) and Triglycerides Assay (OSR60118; Beckman Coulter, Vienna, Austria), respectively. Both analyses were performed on the Olympus AU400 Chemistry Analyzer (Olympus, Tokyo, Japan) in a commercial laboratory (VetLab, Belgrade, Serbia).

### 4.4. Assessment of Insulin Sensitivity

The ipGTT was performed three days before the end of the experimental period. Food was removed 6 h before the test and glucose challenge was given intraperitoneally (2 g/kg), without anesthesia. Blood samples were taken from the tip of the tail at 0, 15, 30, 60, 90, and 120 min after injection, and glucose levels were measured by Accu-Chek strips (Roche, Mannheim, Germany). The AUC that describes glucose concentration as a function of time was calculated by the trapezoidal rule (GraphPad Prism 8, San Diego, CA, USA). HOMA index was calculated from fasting glucose and insulin concentrations, using the formula: [insulin (mIU/L) × glucose (mmol/L)]/22.5.

### 4.5. Histological and Morphometric Analysis

Parts of adipose tissues were fixed in 4% paraformaldehyde for 24 h, than dehydrated in an ethanol gradient, cleared in xylene, and embedded in paraffin. VAT and SAT blocks were sectioned at 7 or 10 µm thickness, respectively, and stained with hematoxylin and eosin. Adipocyte area and diameter were determined with ImageJ plugin Adiposoft 1.16, using manual mode, with calibration based on scale bar measurements, from un-cropped 20× images. Each output was reviewed and counts such as damaged cells or structures resembling cells were excluded. The images for the analysis were obtained using a light microscope Leitz DMRB equipped with a Leica MC190 HD camera and Leica Application Suite (LAS) 4.11.0 software (Leica Microsystems, Wetzlar, Germany) at 20× magnification. The analysis was performed in a blind manner, on 100 cells from high-resolution, randomly located images per section, three sections per animal and six animals per group.

### 4.6. RNA Isolation, Reverse Transcription and Real-Time PCR

Total RNA was extracted from VAT and SAT using TRIzol Reagent (Invitrogen, Waltham, MA, USA), according to the manufacturer’s instructions. Isolated RNA was dissolved in DEPC-treated water and its concentration and purity were determined spectrophotometrically at 260 nm (Nano Photometer N60, Implen GmbH, Munich, Germany). An absorbance ratio of 260/280 that was higher than 1.8 was considered satisfactory.

Complementary DNA (cDNA) was generated from 2 µg of total RNA using the High Capacity Reverse Transcription Kit (Applied Biosystems, Foster City, CA, USA) following the manufacturer’s instructions. The cDNA was stored at −70 °C until use.

TaqMan gene expression FAM labeled probe sets (Applied Biosystems Assay-on-Demand Gene Expression Products, Foster City, CA, USA) were used to determine the expression of LPL (*Lpl*, Rn00561482_m1), and HSL (*Lipe*, Rn00563444_m1), PEPCK (*Pck1*, Rn00563444_m1), ACC (*Acaca*, Rn00573474_m1) in visceral and subcutaneous adipose tissue. TATA box binding protein (*Tbp*, Rn01455646_m1) or β2 microglobulin (*B2m*, Rn00560865_m1) was used as reference genes. The expression of SCD1 (*Scd1*, F: 5′-TGG TGC TCT TTC CCT GTT TGC-3′; R: 5′-TGG GCT TTG GAA GGT GGA CA-3′, Microsynth, Balgach, Switzerland) and FAS (*Fasn*, F: 5′-TTC CTC TGG GAT GTA CCC TCT A-3′; R: 5′-CCG AGT GAA TGA GCA CAG TTT-3′, Microsynth, Balgach, Switzerland) was analyzed by SYBR^®^ Green qPCR, with hypoxanthine guanine phosphoribosyl transferase (*Hprt*, F: 5′-CAG TCC CAG CGT GAT TA-3′; R: 5′-AGC AAG TCT TTC AGT CCT GTC-3′, Invitrogen, Waltham, MA, USA) as a reference gene. All real-time PCR reactions were performed in total volume of 10 µL, using QuantStudio™ Real-Time PCR System (Applied Biosystems, Foster City, CA, USA). The TaqMan reaction mix consisted of 1 × TaqMan^®^ universal PCR master mix, with AmpErase UNG, 1 × TaqMan^®^ gene expression assay, and cDNA template (30 ng of RNA converted to cDNA). SYBR^®^ Green reaction mix contained 1 × power SYBR^®^ Green PCR master mix, specific primer sets, and cDNA template. Thermal cycling conditions were: 2 min incubation at 50 °C, 10 min at 95 °C, followed by 40 cycles at 95 °C for 15 s and 60 °C for 60 s. For SYBR^®^ Green method, melting curve analyses were performed at the end of each experiment to confirm the formation of a single PCR product. To exclude possible reagent contamination, no template controls were used for each target gene. Relative gene expression was calculated using the comparative 2^−∆∆Ct^ method.

### 4.7. Preparation of Cytoplasmic, Nuclear and Mitochondrial Extracts from the VAT

To prepare subcellular protein fractions, frozen samples of VAT were homogenized in 1:1 ratio (*w*/*v*) in ice-cold homogenization buffer (20 mM Tris–HCl, pH 7.2, 10% glycerol, 50 mM NaCl, 1 mM EDTA-Na_2_, 1 mM EGTA-Na_2_, protease inhibitors (2 mM DTT, 20 mM Na_2_MoO_4_, 0.1 mM PMSF) and phosphatase inhibitors (20 mM β-glycerophosphate, 5 mM Na_4_P_2_O_7_, 25 mM NaF)). The homogenates were filtered through gauze and centrifuged at 2000× *g*, 15 min, 4 °C. The resulting pellet was further processed to obtain nuclear fraction, and supernatant was centrifuged at 10,000× *g*, 25 min, 4 °C, to separate mitochondrial fraction, and again at 150,000× *g*, 90 min, 4 °C, to discard microsomal fraction, while the final supernatant was aliquoted and used as cytoplasmic fraction. To obtain nuclear fraction, pellet from initial low-speed centrifugation was washed twice in 1 mL of HEPES buffer (25 mM HEPES, pH 7.6, 10% glycerol, 50 mM NaCl, 1 mM EDTA-Na_2_, 1 mM EGTA-Na_2_, protease inhibitors (2 mM DTT, 20 mM Na_2_MoO_4_, 0.1 mM PMSF) and phosphatase inhibitors (20 mM β-glycerophosphate, 5 mM Na_4_P_2_O_7_, 25 mM NaF)) by centrifugation at 4000× *g*, 10 min, 4 °C. Pellet acquired by washing was resuspended (*w*/*v* = 1:1) in NUN buffer (25 mM HEPES, pH 7.6, 1 M urea, 300 mM NaCl, 1% Nonidet P-40, protease inhibitors (2 mM DTT, 20 mM Na_2_MoO_4_, 0.1 mM PMSF) and phosphatase inhibitors (20 mM β-glycerophosphate, 5 mM Na_4_P_2_O_7_ × 10H_2_O, 25 mM NaF)) and incubated on ice for 90 min, with constant agitation and frequent vortexing. The lysate obtained after incubation was centrifuged at 14,000× *g*, 10 min, 4 °C and final supernatant was aliquoted and used as nuclear fraction.

### 4.8. Preparation of Total Protein from the SAT

The organic phase from total RNA extraction was used to obtain total protein fraction. Total proteins were precipitated from the phenol-ethanol supernatant after addition of isopropanol and subsequent centrifugation at 14,000× *g*, 15 min, 4 °C. The obtained protein pellet was resuspended in 0.3 M guanidine hydrochloride in 95% ethanol and subsequently washed three times in the same buffer. After protein pelleting by the centrifugation at 8000× *g*, 5 min, 4 °C, pellets were dissolved in the 1% SDS and stored at −70 °C until further use.

### 4.9. Western Blot Analysis

Concentration of proteins isolated from VAT and SAT were determined by the Lowry method [45], using bovine serum albumin as a standard. The samples were boiled in 2× Laemmli buffer for 5 min and 30 to 50 µg of proteins were resolved along with a prestained protein ladder (10–170 kDa, Thermo Scientific, Waltham, MA, USA) on 8% or 10% sodium dodecyl sulphate-polyacrylamide gels using Mini-Protean II Electrophoresis Cell system (Bio-Rad Laboratories, Hercules, CA, USA). The proteins were transferred to polyvinylidene difluoride (PVDF) membranes (Immobilon-FL, Millipore, Billerica, MA, USA) in transfer buffer (25 mM Tris, 192 mM glycine, 20% methanol *v*/*v*), at 135 mA, 4 °C, overnight, in Mini Trans-Blot Electrophoretic Transfer Cell system (Bio-Rad Laboratories, Hercules, CA, USA). The membranes were blocked in 5% BSA or non-fat milk for 90 min at room temperature and probed overnight at 4 °C with respective primary antibodies: pIRS-1-Ser^307^ (1:500, Abcam, Cambridge, UK, ab5599), IRS-1 (1:1000, Cell Signaling, Danvers, MA, USA, 2382s), pAkt1/2/3-Ser^473^ (1:500, Santa Cruz Biotechnology, Dallas, TX, USA, sc-514032), Akt1/2/3 (1:500, Santa Cruz Biotechnology, Dallas, TX, USA, sc-81434), pERK1/2-Thr^202^/Tyr^204^ (1:1000, Cell Signaling, Danvers, MA, 9101s), ERK1/2 (1:1000, Cell Signaling, Danvers, MA, 9102s), pAMPKα-Thr^172^ (1:1000, Cell Signaling, Danvers, MA, 2535s), AMPKα1/2 (1:1000, Santa Cruz Biotechnology, Dallas, TX, USA sc-25792). β-actin (1:10,000, Abcam, Cambridge, UK, ab8227), and lamin B1 (1:500, Santa Cruz Biotechnology, Dallas, TX, USA, sc-374015) were used as controls of equal loading. Membranes were extensively washed in PBS containing 0.1% Tween-20 and incubated for 90 min with corresponding mouse (1:30,000, Abcam, Cambridge, UK, ab97046) or rabbit (1:20,000, Abcam, ab6721) HRP-conjugated secondary antibodies. The immunoreactive bands were visualized by enhanced chemiluminescence method using iBright FL1500 Imaging System and quantitative analysis was carried out using iBright Analysis Software (Thermo Fisher Scientific, Waltham, MA, USA).

### 4.10. Statistical Analyses

All data are given as mean ± standard error of the mean (SEM). *Two-way* repeated measures ANOVA followed by Sidak’s *post hoc* test was used to compare masses of animals from small and normal litters during the weaning period. To determine the effects of litter size reduction and DHT treatment, as well as their interaction, a *two-way* analysis of variance (ANOVA) was used. When there was significant interaction between factors, inter-group differences were analyzed by the Tukey *post hoc* test. The differences between groups were considered significant at *p* < 0.05. Statistical analyses were performed using STATISTCA 7.0 (StatSoft Inc., Tulsa, OK, USA) and GraphPad Prism 8 (San Diego, CA, USA) software.

## Figures and Tables

**Figure 1 ijms-23-08942-f001:**
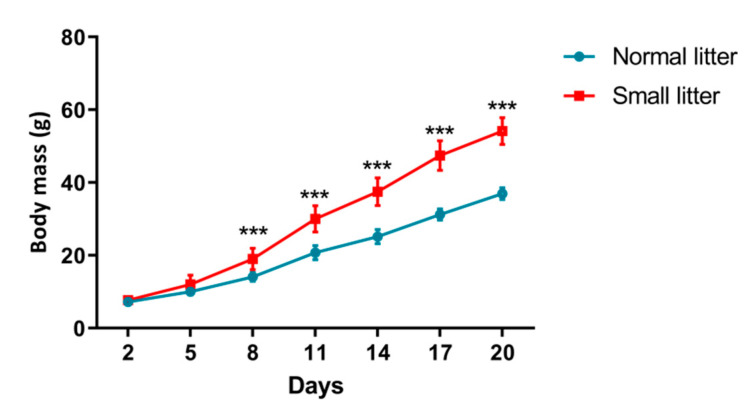
Increase in body mass in animals from normal vs. small litters during the suckling period. Each point represents the mean ± standard deviation. Comparison between masses of animals (*n* = 12 animals per group) from small and normal litters by time was done by *two-way* repeated measures ANOVA followed by Sidak’s *post hoc* test and the difference is considered significant at *p* < 0.05. Symbols denote significant difference in masses between normal litter (NL) raised and small litter (SL) raised animals for certain time points (*** *p* < 0.001).

**Figure 2 ijms-23-08942-f002:**
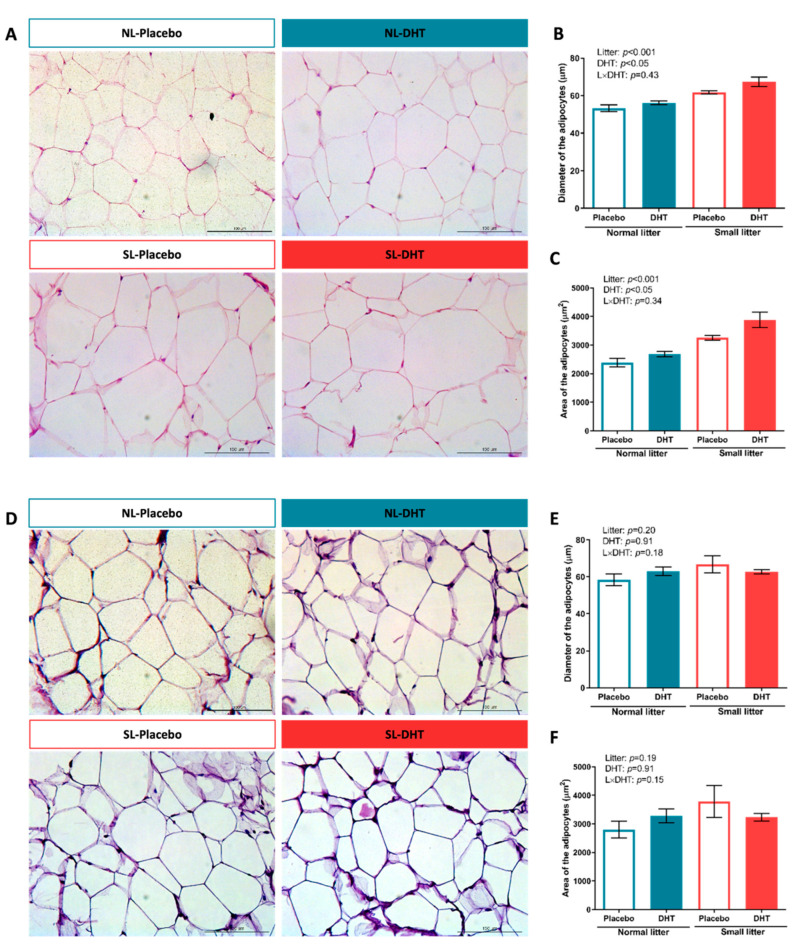
Histological and morphometric analysis of VAT and SAT of placebo and DHT treated rats from small litters (SL) and normal litters (NL). Representative micrographs of hematoxylin-eosin stained sections of (**A**) VAT and (**D**) SAT (bar = 100µm), and morphometric analysis of (**B**,**E**) cell diameter and (**C**,**F**) cell area in the VAT and SAT, respectively. Data are shown as mean ± SEM (*n* = 6 animals per group). To determine the effects of litter size reduction and DHT treatment, as well as their interaction, a *two-way* ANOVA was performed, and the difference was considered significant at *p* < 0.05.

**Figure 3 ijms-23-08942-f003:**
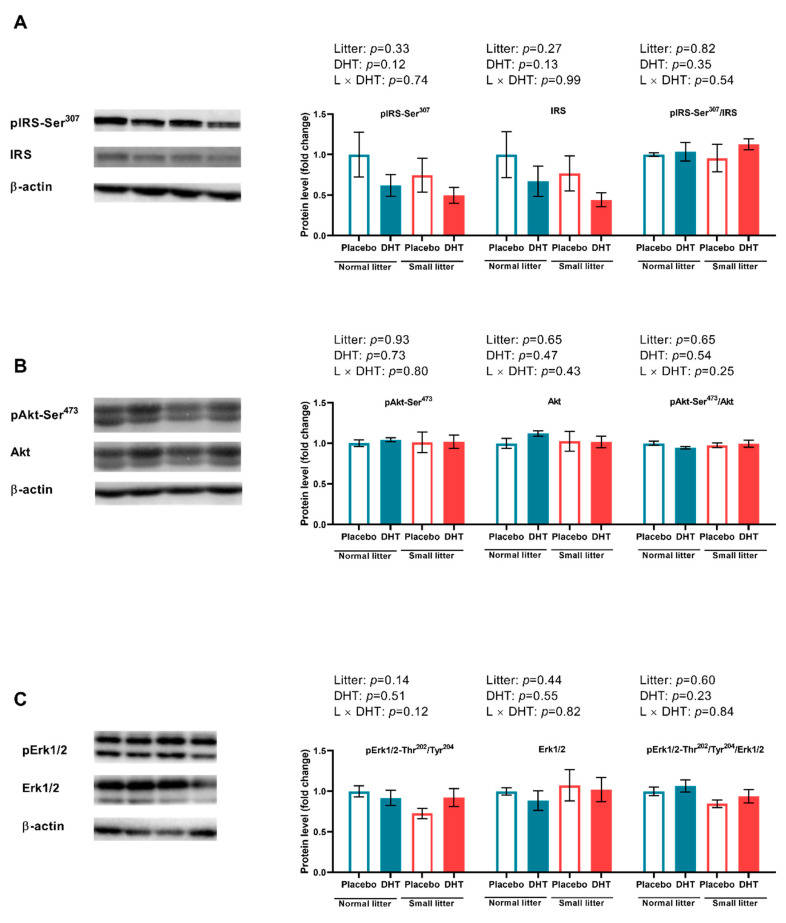
Insulin signaling in VAT of placebo and DHT treated rats from small and normal litters. (**A**) Protein levels of pIRS1-Ser^307^, IRS1 and their relative ratio with representative Western blots; (**B**) Protein levels of pAkt-Ser^473^, Akt and their relative ratio with representative Western blots; (**C**) Protein levels of pErk1/2-Thr^202^/Tyr^204^, Erk1/2 and their relative ratio with representative Western blots. All protein levels were measured in cytosol of the VAT and normalized to β actin. The data are presented as mean ± SEM (*n* = 6 animals per group). To determine the effects of litter size reduction and DHT treatment, as well as their interaction, *two-way* ANOVA was done, and the difference was considered significant at *p* < 0.05.

**Figure 4 ijms-23-08942-f004:**
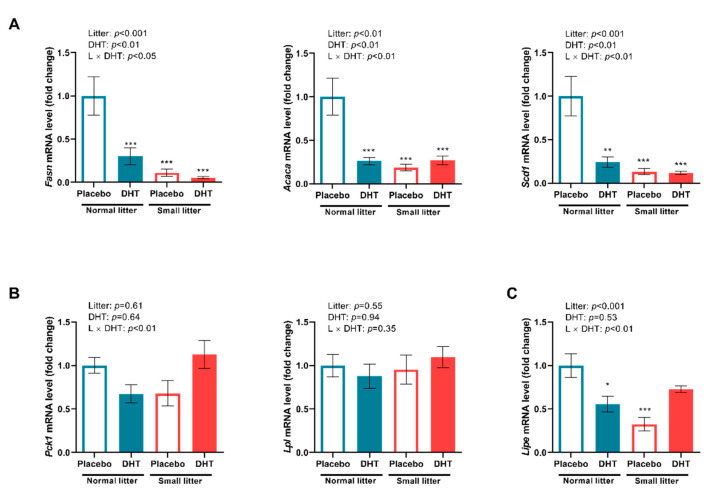
Relative expression of lipogenesis and lipolysis markers in the VAT of placebo and DHT treated rats from small and normal litters. (**A**) Relative expression of *Fasn*, *Acaca,* and *Scd1* gene. (**B**) Relative expression of *Pck1* and *Lpl* gene. (**C**) Relative expression of *Lipe* gene. The gene expression was normalized to *Hprt*, *Tbp* or *B2m* gene expression and the data are presented as mean ± SEM (*n* = 6 animals per group). To determine the effects of litter size reduction and DHT treatment, as well as their interaction, *two-way* ANOVA, followed by Tukey *post hoc* test was performed, and the difference was considered significant at *p* < 0.05. Symbols denote significant statistical differences between NL-Placebo and NL-DHT, SL-Placebo or SL-DHT groups (* *p* < 0.05, ** *p* < 0.01, *** *p* < 0.001).

**Figure 5 ijms-23-08942-f005:**
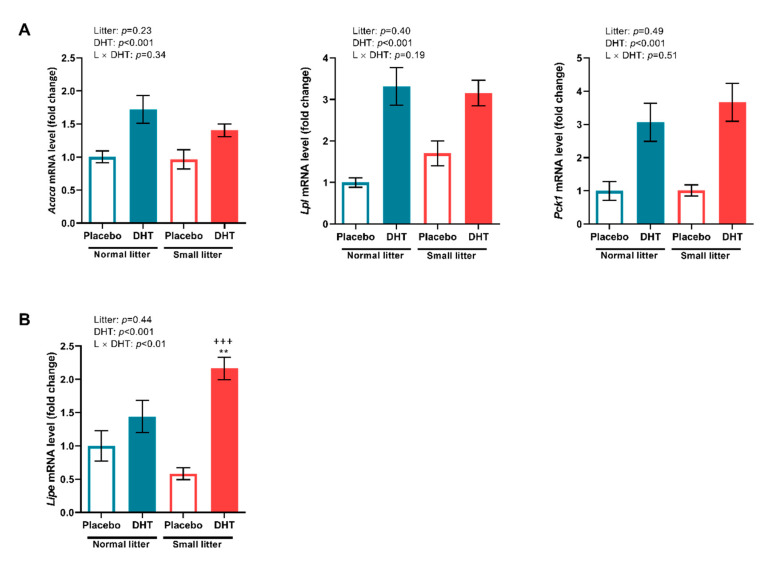
Relative expression of lipogenesis and lipolysis markers in the SAT of placebo and DHT treated rats from small and normal litters. (**A**) Relative expression of *Acaca*, *Lpl and Pck1* genes. (**B**) Relative expression of *Lipe* gene. The expression of the genes was normalized to the expression of *Tbp* or *B2m* gene and the data are presented as mean ± SEM (*n* = 6 animals per group). To determine the effects of litter size reduction and DHT treatment, as well as their interaction, *two-way* ANOVA, followed by Tukey *post hoc* test was performed, and the difference was considered significant at *p* < 0.05. Symbols denote a statistically significant difference between NL-Placebo and SL-DHT groups (** *p* < 0.01), or between SL-Placebo and SL-DHT groups (^+++^
*p* < 0.001).

**Figure 6 ijms-23-08942-f006:**
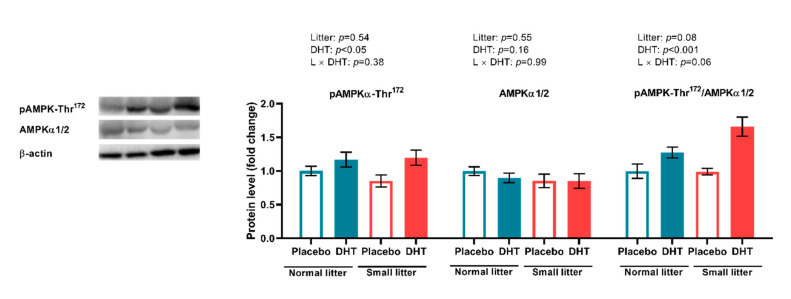
The protein levels of phosphorylated and total AMPK, and their relative ratio in the VAT of placebo and DHT treated rats from small and normal litters. All protein levels were measured in cytosolic fraction of VAT, normalized to β actin. Representative Western blots are shown. The data are presented as mean ± SEM (*n* = 6 animals per group). To determine the effects of litter size reduction and DHT treatment, as well as their interaction, *two-way* ANOVA, was performed, and the difference was considered significant at *p* < 0.05.

**Figure 7 ijms-23-08942-f007:**
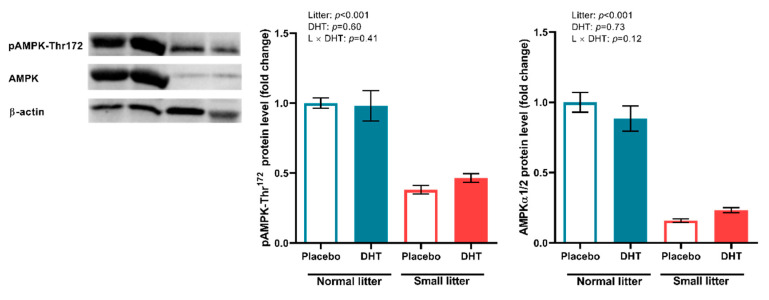
The protein levels of phosphorylated and total AMPK in the SAT of placebo and DHT treated rats from small and normal litters. All protein levels were measured in total protein extract, normalized to β actin. Representative Western blots are shown. The data are presented as mean ± SEM (*n* = 6 animals per group). To determine the effects of litter size reduction and DHT treatment, as well as their interaction, *two-way* ANOVA was performed, and the difference was considered significant at *p* < 0.05.

**Table 1 ijms-23-08942-t001:** Energy intake, body mass and absolute and relative masses of the visceral and subcutaneous adipose tissue (VAT and SAT, respectively) and reproductive organs of DHT treated and placebo rats from small litters (SL) and normal litters (NL).

	NL-Placebo	NL-DHT	SL-Placebo	SL-DHT	Two-Way ANOVA		
					Litter	DHT	Interaction
**Energy intake (kJ/day/cage)**	475.5 ± 10	467.6 ± 15.5	512.4 ± 13.6	531.1 ± 13.4	<0.001	ns	ns
**Body mass (g)**	230.0 ± 4.6	241.0 ± 11.0	252.33 ± 4.2	283.33 ± 13.0	<0.001	<0.05	ns
**Mass of VAT (g)**	7.0 ± 0.6	8.4 ± 1.4	14.0 ± 1.4	12.0 ± 1.1	<0.001	ns	ns
**VAT-to-body ratio (×100)**	3.0 ± 0.2	3.3 ± 0.4	5.3 ± 0.5	4.5 ± 0.4	<0.001	ns	ns
**Mass of SAT (g)**	2.1 ± 0.3	1.8 ± 0.4	2.0 ± 0.17	2.2 ± 0.3	ns	ns	ns
**SAT-to-body ratio (×100)**	0.9 ± 0.1	0.7 ± 0.1	0.8 ± 0.1	0.8 ± 0.1	ns	ns	ns
**Mass of ovary (g)**	0.048 ± 0.003	0.027 ± 0.006	0.043 ± 0.002	0.025 ± 0.004	ns	<0.001	ns
**Ovary-to-body ratio (×1000)**	0.21 ± 0.01	0.11 ± 0.03	0.17 ± 0.01	0.085 ± 0.01	ns	<0.001	ns
**Mass of uterus (g)**	0.49 ± 0.06	0.2 ± 0.03	0.39 ± 0.05	0.24 ± 0.04	ns	<0.001	ns
**Uterus-to-body ratio (×1000)**	2.1 ± 0.22	0.85 ± 0.14	1.6 ± 0.22	0.86 ± 0.16	ns	<0.001	ns

All data are presented as mean ± standard error of mean (SEM) (*n* = 6 animals per group). To determine the effects of litter size reduction and DHT treatment, as well as their interaction, *two-way* ANOVA was performed, and the difference was considered significant at *p* < 0.05; ns—non-significant.

**Table 2 ijms-23-08942-t002:** Lipid status and systemic insulin sensitivity parameters of DHT treated and placebo rats from small litters (SL) and normal litters (NL).

	NL-Placebo	NL-DHT	SL-Placebo	SL-DHT	Two-Way ANOVA		
					Litter	DHT	Interaction
**Triglycerides (mmol/L)**	0.87 ± 0.09	0.88 ± 0.11	0.73 ± 0.03	0.93 ± 0.07	ns	ns	ns
**FFA (mmol/L)**	1.53 ± 0.29	1.22 ± 0.18	1.15 ± 0.21	1.20 ± 0.25	ns	ns	ns
**Glucose (mmol/L)**	6.00 ± 0.18	5.82 ± 0.14	5.85 ± 0.14	5.83 ± 0.16	ns	ns	ns
**Insulin (µU/mL)**	17.0 ± 0.7	19.0 ± 2.7	23.32 ± 1.2	36.0 ± 3.9 ***^,###,††^	<0.001	<0.01	<0.05
**HOMA index**	4.5 ± 0.1	4.9 ± 0.6	6.1 ± 0.3	9.3 ± 1.1 ***^,###,††^	<0.001	<0.05	<0.05
**ipGTT AUC**	1249 ±49.1	1266 ± 90.0	1442 ± 45.4	1578 ± 51.01	<0.001	ns	ns

All data are presented as mean ± SEM (*n* = 5–6 animals per group). To determine the effects of litter size reduction and DHT treatment, as well as their interaction, a *two-way* ANOVA, followed by the Tukey *post hoc* test was performed, and the difference was considered significant at *p* < 0.05. Symbols denote significant differences from NL-Placebo group (*** *p* < 0.001); NL-DHT group (^###^
*p* < 0.001) or SL-Placebo group (^++^
*p* < 0.01); ns—non-significant.

## Data Availability

The data presented in this study are available on request from the corresponding author.

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
