# Peer review of "AMPK Activation Is Important for the Preservation of Insulin Sensitivity in Visceral, but Not in Subcutaneous Adipose Tissue of Postnatally Overfed Rat Model of Polycystic Ovary Syndrome"

_ijms, 2022, doi:10.3390/ijms23168942_

Round 1
Reviewer 1 Report
I believe that the authors have done a exhaustive and well-organized study with interesting results. I congratulate the authors for that.
Reviewer 2 Report
The study presents a detailed analysis of visceral and subcutaneous adipose tissue in response to a PCOS animal model. The results of the study are of interest. Some of the results of the 2-way ANOVAs are misinterpreted (see specific comments below); therefore, revision is required in some part of the results and discussion sections.
I suggest revising the title so that the term polycystic ovarian syndrome appears somewhere in the title
Line 27 change the word taking to taken
Line 59 delete the word that
Line 85 change to AMPK is an energy sensor of the cell…
Lines 88 to 91 some references are required here to support the statements about AMPK
Please add a hypothesis statement at the end of the introduction
Unless it is the journal style I suggest having the method section before the results section
Early in the results section please indicate the type of animal that was used in the experiments
Line 464 change the word cloth to clot
Line 484 define the abbreviation IPGTT
Please indicate if any of the assessments were done in a blinded manner, that is, did any of the assessors know which groups the animals were assigned to?
Line 516 and 517 please define the abbreviations LPL and HSL
Figure 1: instead of using unpaired students T tests to analyze this data you should use a group by time and anova with Post Hoc tests
Table 1: Please check that there is actually a main effect of DHT for mass of VAT. When looking at the means for the placebo groups combined versus the DHT groups combined, these means are very similar, so I am not sure there is a DHT main effect here.
Table 1: Here you present results for the main effects of litter, DHT, and the interaction between litter and DHT. The last three columns of the table are very clear in the presentation of the statistical results. Within the table however, you have presented differences between the individual groups, which I don’t think you can do unless you are conducting a post-hoc test on a significant interaction. You don’t have any significant interactions, so I don’t think you can present these results. Presenting the litter and DHT main effects (in the last three columns) is adequate.
I have a similar comment for table 2: Here it is appropriate to present differences between individual groups in the table for Insulin and HOMA because you have an interaction and can therefore conduct the post-hoc tests. You do not however have an interaction for ipGTT AUC and therefore should not be doing the post-hoc test to determine differences between the individual groups. Here it is appropriate just to indicate there was a main effect due to litter size (this indicates the SL groups combined were greater than the NL groups combined).
Line 186: Figure 2: Here you indicate you had main effects for litter size and DHT, but you have presented differences between individual groups in your figures, which would be the result of a post-hoc test on a significant interaction. Again, please indicate whether the interaction is significant. If it is, then the post-hoc tests are appropriate, but if it is not significant, then please just present the main effects from your 2-way ANOVA.
Line 253: Here you indicate you had a main effect due to DHT (Figure 5A), but again you have presented differences between individual groups as if you have a litter x DHT interaction. Again, please only present the DHT main effect here. In the graph, you can show that the DHT groups combined were greater than the placebo groups combined.
Figure 6: Same comment: You have a DHT main effect. You can show in the graph that the combined DHT groups were greater than the combined placebo groups, but you shouldn’t be showing differences between individual groups because you don’t appear to have an interaction.
Same comment for Figure 7. Please present the main effect for litter size (combined normal litter groups > combined small litter groups) instead of the post-hoc results of a non-significant interaction.
Line 329: Change the word “besides” to “also”.
Lines 343-345: “In the present study, combination of postnatal overfeeding and DHT treatment have conditioned an increase in blood insulin levels and HOMA index, while blood glucose levels remained unchanged.” I don’t think the last part of this sentence is accurate because litter size (i.e. overfeeding) increased ipGTT AUC.
Lines 345-347: “The contribution of postnatal overfeeding to metabolic features of PCOS is evidenced by ipGTT that demonstrated decreased insulin sensitivity only after combined treatments.” I don’t think this statement is accurate, as there was only a litter main effect (i.e. an effect due to overfeeding) and no effect of DHT.
Line 419: “. On the other hand, in the SAT of the animals exposed to combined treatment, activity of AMPK was decreased, while the simultaneous increase in lipogenic and lipolytic markers maintained SAT adipocytes morphology unchanged” – I don’t think this statement is accurate because you only had a main effect due to litter for AMPK (small litter and therefore overfeeding decreased AMPK, but DHT had no effect). Also, you only had a main effect for DHT for genes related to lipogenesis and HSL (i.e. these were upregulated in the combined DHT groups vs. the combined placebo groups).
Round 2
Reviewer 2 Report
The manuscript is improved and results are now more clearly presented and interpreted
Author Response
Dear Reviewer,
We are grateful to you for the useful suggestions and corrections of the manuscript that have improved its quality. We hope that the revised version of the manuscript is suitable for publication in the Topical Collection entitled “Recent Advances in Molecular Research of Metabolic Disorders” of The International Journal of Molecular Sciences.
Best regards,
Danijela Vojnović Milutinović